# DISCRETE PLANNING WITH NEURO-ALGORITHMIC POLICIES

## ABSTRACT

Although model-based and model-free approaches to learning the control of systems have achieved impressive results on standard benchmarks, generalization to variations in the task are still unsatisfactory. Recent results suggest that generalization of standard architectures improves only after obtaining exhaustive amounts of data. We give evidence that the generalization capabilities are in many cases bottlenecked by the inability to generalize on the combinatorial aspects. Further, we show that for a certain subclass of the MDP framework, this can be alleviated by neuro-agorithmic architectures.

Many control problems require long-term planning that is hard to solve generically with neural networks alone. We introduce a neuro-algorithmic policy architecture consisting of a neural network and an embedded time-depended shortest path solver. These policies can be trained end-to-end by blackbox differentiation. We show that this type of architecture generalizes well to unseen variations in the environment already after seeing a few examples.

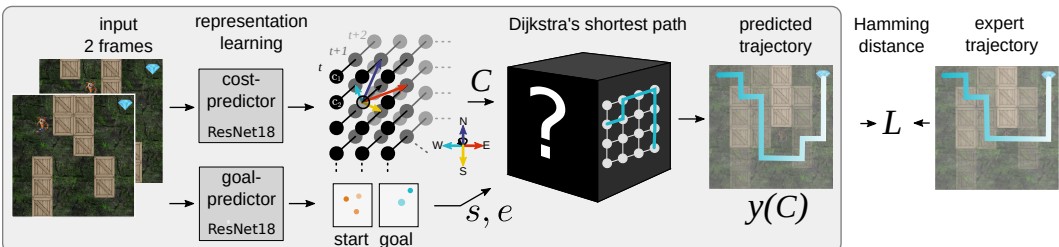

Figure 1: Architecture of the neuro-algorithmic policy. Two subsequent frames are processed by two simplified ResNet18s: the cost-predictor outputs a tensor (width $\times$ height $\times$ time) of vertex costs $c_t^v$ and the goal-predictor outputs heatmaps for start and goal. The time-dependent shortest path solver finds the shortest path to the goal. Hamming distance between the proposed and expert trajectory is used as loss for training.

## 1 INTRODUCTION

One of the central topics in machine learning research is learning control policies for autonomous agents. Many different problem settings exist within this area. On one end of the spectrum are imitation learning approaches, where prior expert data is available and the problem becomes a supervised learning problem. On the other end of the spectrum lie approaches that require interaction with the environment to obtain data for policy extraction problem, also known as the problem of exploration. Most Reinforcement Learning (RL) algorithms fall into the latter category. In this work, we concern ourselves primarily with the setting where limited expert data is available, and a policy needs to be extracted by imitation learning.

Independently of how a policy is extracted, a central question of interest is: how well will it generalize to variations in the environment and the task? Recent studies have shown that standard deep RL algorithms require exhaustive amounts of exposure to environmental variability before starting to generalize Cobbe et al. (2019).

There exist several approaches addressing the problem of generalization in control. One option is to employ model-based approaches that learn a transition model from data and use planning algorithms at runtime. This has been argued to be the best strategy in the presence of an accurate model and sufficient computation time (Daw et al., 2005). However, learning a precise transition model is often harder than learning a policy. This, in turn, makes them more general, but comes at a cost of increasing the problem dimensionality. The transition model has a much larger dimensionality and it needs to model aspects of the environmental dynamics that are perhaps irrelevant for the task. This is particularly true for learning in problems with high-dimensional inputs, such as raw images. In order to alleviate this problem, learning specialized or partial models has shown to be a viable alternative, e.g. in MuZero Schrittwieser et al. (2019).

We propose to use recent advances in making combinatorial algorithms differentiable in a blackbox fashion as proposed by Vlastelica et al. (2020) to train neuro-algorithmic policies with embedded planners end-to-end. More specifically, we use a time-dependent shortest path planner acting on a temporally evolving graph generated by a deep network from the inputs. This enables us to learn the time-evolving costs of the graph and relates us to model-based approaches. We demonstrate the effectiveness of this approach in an offline imitation learning setting, where a few expert trajectories are provided. Due to the combinatorial generalization capabilities of planners, our learned policy is able to generalize to new variations in the environment out of the box and orders of magnitude faster than naive learners. Using neuro-algorithmic architectures facilitates generalization by shifting the combinatorial aspect of the problem to efficient algorithms, while using neural networks to extract a good representation for the problem at hand. They have potential to endow artificial agents with the main component of intelligence, the ability to reason.

Our contributions can be summarized as follows:

- We identify that poor generalization is caused by lack of structural and combinatorial inductive biases and can be alleviated by introducing the correct inductive biases through neuro-algorithmic policies.
- We show that architectures embedding TDSP solvers are applicable beyond goal-reaching environments.
- We demonstrate learning neuro-algorithmic policies in dynamic game environments from images.

## 2    RELATED WORK

**Planning**    There exist multiple lines of work aiming to improve classical planning algorithms such as improving sampling strategies of Rapidly-exploring Random Trees (Gammell et al., 2014; Burget et al., 2016; Kuo et al., 2018). Similarly, along this direction, Kumar et al. (2019) propose a conditional VAE architecture for sampling candidate waypoints. Orthogonal to this are approaches that learn representations such that planning is applicable in the latent space. Hafner et al. (2019) employ a latent multi-step transition model. Savinov et al. (2018) propose a semi-parametric method for mapping observations to graph nodes and then applying a shortest path algorithm. Asai & Fukunaga (2017); Asai & Kajino (2019) use an autoencoder architecture in order to learn a discrete transition model suitable for classical planning algorithms. Li et al. (2020) learn compositional Koopman operators with graph neural networks mapping to a linear dynamics latent space, which allows for fast planning. Chen et al. (2018); Amos et al. (2017) perform efficient planning by using a convex model formulation and convex optimization. Alternatively, the replay buffer can be used as a non-parametric model in order to select waypoints (Eysenbach et al., 2019) or in an MPC fashion (Blundell et al., 2016). None of these methods perform differentiation through the planning algorithm in order to learn better latent representations.

**Differentiation through planning**    Embedding differentiable planners has been proposed in previous works, e.g. in the continuous case with CEM (Amos & Yarats, 2020; Bharadhwaj et al., 2020). Wu et al. (2020) use a (differentiable) recurrent neural network as a planner. Tamar et al. (2016) use a differentiable approximation of the value iteration algorithm to embed it in a neural network. Silver et al. (2017b) differentiate through a few steps of value prediction in a learned MDP to match the externally observed rewards. Srinivas et al. (2018) use a differentiable forward dynamics model in latent space. Karkus et al. (2019) suggest a neural network architecture embedding MDP and POMDP

solvers and during the backward pass, they substitute the algorithms by learned approximations. In comparison, we do not perform any relaxation or approximation of the planner itself and we learn interpretable time-dependent costs of the latent planning graph based on expert demonstrations by differentiating through the planner. Similarly to our work, Yonetani et al. (2020) embed an A$^*$ algorithm into a neural network, but in comparison, their method does not operate with time-dependent costs, subgoal selection and does not provide a policy for closed-loop control.

**Inverse reinforcement learning and imitation learning**   Uncovering the expert objective function from demonstrations has been a central topic in reinforcement learning (Ng & Russell, 2000). Our method is connected to inverse reinforcement learning in the sense that we learn the objective function that the expert optimizes to extract an optimal policy, also called apprenticeship learning (Abbeel & Ng, 2004; Neu & Szepesvári, 2012; Aghasadeghi & Bretl, 2011). What separates our approach is that the inferred costs are inherently part of the learned neuro-algorithmic policy in conjunction with the applied planner on the costs.

Our method is an offline imitation learning method, but since we propose an end-to-end trainable policy, it is naturally extendable to the online case with a method such as DAgger (Ross et al., 2011) or other online reinforcement learning methods augmented with expert datasets (Reddy et al., 2019; Ho & Ermon, 2016).

**Offline model-based reinforcement learning**   Model-based methods have shown promise by facilitating better generalization (Janner et al., 2019). Approaches employing models fall into two camps: using models to extract a policy in a Dyna-style approach (Sutton, 1991; Janner et al., 2019; Sutton et al., 2008; Yao et al., 2009; Kaiser et al., 2019), or incorporating the model in a planning loop, i.e. model-predictive control (Finn & Levine, 2017; Racanière et al., 2017; Oh et al., 2017; Silver et al., 2017a). In this work, we consider the latter case where an implicit transition model is "hidden" within the predicted time-dependent costs.

**Combinatorial algorithms in end-to-end trainable networks**   We suggest a hybrid policy consisting of a neural network and an accompanying expert (shortest path) discrete solver that is trainable end-to-end. Incorporating expert discrete solvers into end-to-end trainable architectures is a topic with exciting recent developments. For the simpler setup of comparing to ground-truth values on the solver output, numerous frameworks have been suggested such as the "predict-and-optimize" framework and its variants (Elmachtoub & Grigas, 2017; Demirovic et al., 2019; Mandi et al., 2019). Also, specializations for concrete cases such as sparse structured inference (Niculae et al., 2018), logical satisfiability (Wang et al., 2019), submodular optimization (Djolonga & Krause, 2017) or mixed integer programming (Ferber et al., 2020) have been proposed.

We are interested in the harder case of providing an *entirely hybrid* architecture which may use the solver at intermediate levels and is trainable end-to-end. For this case, two approaches have recently emerged (Vlastelica et al., 2020; Berthet et al., 2020). Vlastelica et al. (2020) introduce an efficient implicit piece-wise linear interpolation scheme, while Berthet et al. (2020) introduce Monte Carlo technique for estimating the Jacobian of a Gaussian smoothing of the piecewise constant function. The approach from Vlastelica et al. (2020) is especially appealing, since it allows for uses in which the solver is the computational bottleneck. Therefore, we follow it in this work. By formulating the control problem as a time-dependent shortest path problem (TDSP), we show that the framework from Vlastelica et al. (2020) is applicable in specific control settings.

## 3   MARKOV DECISION PROCESSES AND SHORTEST PATHS

We follow the MDP framework Puterman (2014) in a goal-conditioned setting Schaul et al. (2015). This is used in sequential decision making problems where a specific terminal state has to be reached.

**Definition 1** *A goal-conditioned Markov Decision Process (gcMDP), $\mathcal{M}$ is defined by the tuple ($\mathcal{S}$, $\mathcal{A}$, $p$, $g$, $r$), where $\mathcal{S}$ is the state space, $\mathcal{A}$ the action space, $p(s' \mid a, s)$ the probability of making the transition $s \rightarrow s'$ when taking the action $a$, $g$ is the goal, $r(s, a, s', g)$ the reward obtained when transitioning from state $s$ to $s'$ while taking action $a$ and aiming for goal $g$.*

Concretely, we concern ourselves with fully observable discrete MDPs, in which the Markov assumption for the state holds and where the state and action-space are discrete. The goal of reinforcement learning is to maximize the return $G = \sum_{t=0}^{T} r_t$ of such a process. In gcMDPs the reward is such that the maximal return can be achieved by reaching the goal state $g$.

Given access to the transition probabilities and rewards, an optimal policy can be extracted by dynamic programming Bertsekas et al. (1995). In a graph representation of an gcMDP, the set of vertices $V$ corresponds to the set of states $\mathcal{S}$, traversing an edge corresponds to making a transition between states. We assume a deterministic process, such that the optimal policy can be extracted by standard shortest path algorithms, such as Dijkstra's algorithm.

In this work, we imitate expert trajectories by training a policy with an embedded time-dependent shortest path solver end-to-end. Although the actual gcMDP solved by the expert may be stochastic, we learn a deterministic latent approximate gcMDP, $\widehat{\mathcal{M}}$. Assuming that we have access to the topology of the gcMDP, by applying blackbox-differentiation theory Vlastelica et al. (2020) we are able to learn the underlying costs (instead of rewards) of $\widehat{\mathcal{M}}$ such that the optimal policy on $\widehat{\mathcal{M}}$ is also optimal in $\mathcal{M}$.

Although the MDP Definition 1 yields itself nicely towards learning the time-dependent edge costs $c_t^e$, this can increase the problem dimensionality considerably with the out-degree of the vertices (here $|\mathcal{A}|$). Thus, we consider cases where the reward function only depends on the current state and the goal: $r(s, a, s', g) = r(s, g)$. In this case, vertex costs $c_t^v$ are sufficient for finding the optimal solution to the gcMDP. Accordingly, we rely on a vertex-based version of the shortest path algorithm.

## 4 Shortest Path Algorithm and its Differentiation

We will employ an efficient implementation of Dijkstra's algorithm for computing the shortest path. For differentiation, we rely on the framework for blackbox differentiation of combinatorial solvers Vlastelica et al. (2020).

### 4.1 Time-dependent Shortest Path

The purely combinatorial setup can be formalized as follows. Let $G = (V, E)$ be a graph. For every $v_i \in V$, let $c_i^1, \ldots, c_i^T$ be non-negative real numbers; the costs of reaching the vertex $v_i$ at time-points $1, 2, \ldots, T$, where $T$ is the planning horizon. The TIME-DEPENDENT-SHORTEST-PATH problem (TDSP) has as input the graph $G$, a pair of vertices $s, e \in V$ (start and end) and the matrix $C \in \mathbb{R}^{|V| \times T}$ of the costs $c_i^t$. This version of the shortest path problem can be solved by executing the Dijkstra shortest path algorithm[1] Dijkstra (1959) on an augmented graph. In particular, we set

$$V^* = \{(v, t) \colon v \in V, t \in [1, T]\}$$
$$E^* = \{((v_1, t), (v_2, t+1)) \colon (v_1, v_2) \in E^{\hookrightarrow}, t \in [1, T-1]\},$$

where the cost of vertex $(v_i, t) \in V^*$ is simply $c_i^t$ and $E^{\hookrightarrow}$ is the original edge set $E$ appended with all self-loops. This allows to "wait" at a fixed vertex $v$ from timestep $t$ to timestep $t+1$. In this graph, the task is to reach vertex $(e, T)$ from $(s, 1)$ with the minimum traversal cost.

The time-dependent shortest path problem can be used for model predictive control using receding horizon planning, as done in our approach.

### 4.2 Applicability of Blackbox Differentiation

The framework presented in Vlastelica et al. (2020) turns blackbox combinatorial solvers into neural network building blocks. The provided gradient is based on a piecewise linear interpolation of the true piecewise constant (possibly linearized) loss function, see Fig. 2. The *exact* gradient of this linear interpolation is computed efficiently via evaluating the solver on only one more instance (see Algorithm 1).

---

[1]Even though the classical formulation of Dijkstra's algorithm is edge-based, all of its properties hold true also in this vertex based formulation.

---

**Algorithm 1** Forward and backward Pass for the shortest-path algorithm

---

**function** FORWARDPASS($C, s, e$)
    $Y := \textbf{TDSP}(C, s, e)$                                       *// Run Dijkstra's algorithm*
    **save** $Y, C, s, e$                                            *// Needed for backward pass*
    **return** $Y$

**function** BACKWARDPASS($\nabla L(Y), \lambda$)
    **load** $Y, C, s, e$
    $C_\lambda := C + \lambda \nabla L(Y)$                                 *// Calculate modified costs*
    $Y_\lambda := \textbf{TDSP}(C_\lambda, s, e)$                              *// Run Dijkstra's algo.*
    **return** $\frac{1}{\lambda}(Y_\lambda - Y)$

---

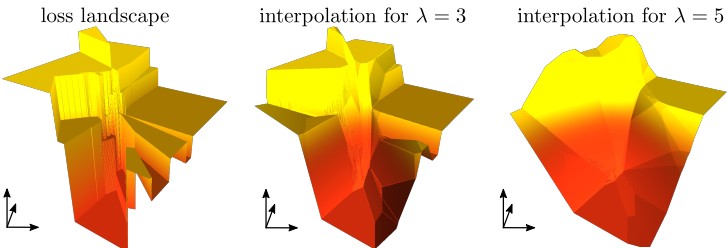

loss landscape        interpolation for $\lambda = 3$        interpolation for $\lambda = 5$

Figure 2: Differentiation of a piecewise constant loss resulting from incorporating a combinatorial solver. A two-dimensional section of the loss landscape is shown (left) along with two differentiable interpolations of increasing strengths (middle and right).

In order to apply this differentiation scheme, the solver at hand needs to have a formulation in which it minimizes an inner-product objective (under arbitrary constraints). To that end, for a given graph $G = (V, E)$ with time-dependent costs $C \in \mathbb{R}^{|V| \times T}$ we define $Y \in \{0, 1\}^{|V| \times T}$ an indicator matrix of visited vertices. In particular, $Y_i^t = 1$ if and only if vertex $v_i$ is visited at time point $t$. The set of such indicator matrices that correspond to valid paths in the graph $(V^*, E^*)$ will be denoted as $\mathrm{Adm}(G)$. The time-dependent shortest path optimization problem can be then rewritten as

$$\mathrm{TDSP}(C, s, e) = \underset{Y \in \mathrm{Adm}(G)}{\arg\min} \sum_{(i,t)} Y_i^t C_i^t. \tag{1}$$

This is an inner-product objective and thus the theory from Vlastelica et al. (2020) applies. In effect, the deep network producing the cost tensor $C$ can be trained via supervision signal from ground truth shortest paths.

### 4.3 COST MARGIN

Our work is related to the problem of metric learning in the sense that we learn the distance metric between the current position of the agent (state) and the target position in the underlying gcMDP, allowing us to solve it with a shortest path algorithm. It has been shown that inducing a margin on the metric can be beneficial for generalization. Similarly to (Rolínek et al., 2020) in the context of rank-based metric learning, we induce a margin $\alpha$ on the costs of the latent gcMDP, increasing the cost of ground truth path and decreasing the rest in the training stage of the algorithm:

$$c_i^t = \begin{cases} c_i^t + \frac{\alpha}{2} & \text{if } (v_i, t) \in Y^* \\ c_i^t - \frac{\alpha}{2} & \text{if } (v_i, t) \notin Y^* \end{cases} \quad \forall \, (v_i, t) \in V^*. \tag{2}$$

During evaluation, the cost margin is removed from the shortest path calculation.

## 5 NEURO-ALGORITHMIC POLICY ARCHITECTURE

We propose the Neuro-algorithmic Policy (NAP) framework, that is an end-to-end trainable deep policy architecture embedding an algorithmic component using the afore-mentioned techniques.

In this paper we consider a concrete architecture consisting of two main components: a backbone ResNet18 (without the final fully connected layers, a detailed description is available in Sec. C of the appendix) architecture and the shortest path solver, see Fig. 1. At each time step the policy receives two images concatenated channel-wise from which it predicts the cost matrix $C$ for the planning horizon $T$ with the *cost-predictor* and the start vertex $s$ and end vertex $e$ with the *goal predictor*, explained below.

The cost matrix $C$ is given to the solver along with the start vertex $s$ and end vertex $e$ to compute the time-dependent shortest path $Y$. The *cost-predictor* is trained using the Hamming distance between the predicted plan $Y$ and the expert plan $Y^*$ that we use for supervision.

The policy is used in a model-predictive control setting, i.e. at execution time we predict the plan $Y$ for horizon $T$ at each time step and execute the first action from the plan.

## 5.1 GOAL PREDICTION, GLOBAL AND LOCAL

In order to apply the solver to the learned latent graph representation, we need to map the current state of the environment to appropriate start and end vertices $(s, e)$. To this end, we employ a second ResNet18 – the *goal-predictor* – similar to the *cost-predictor* that learns to extract the agent start position and a suitable target position. The training of this predictor is using a Cross-Entropy loss and is independent of learning the costs of the latent graph representation.

At training time, given the expert trajectories $Y^*$ we have access to the current position of the agent and its position in the future. Thus, for predicting $s$ we have a simple supervision signal, namely the current position. For the goal prediction $e$ we extract a set of suitable goal locations from the expert. Here, we distinguish between *global* and *local* planning.

In the *global* setting, the last position of the expert is the goal $e$, corresponding to, for instance, the jewel in CRASH JEWEL HUNT, see Fig. 3.

In the *local* setting, we expect the end vertex to be an intermediate goal ("collect an orb"), which effectively allows for high-level planning strategies while the low-level planning is delegated to the discrete solver. In this case, the positively labeled supervision at time $t$ are all locations of the (expert) agent between step $t + T$ and $t + 2T$.

The local setting allows to limit the complexity of our method, which grows with the planning horizon. This is also a trade-off between the combinatorial complexity solved by the TDSP solver and the goal predictor. Ideally, the planning horizon $T$ used for the cost-prediction is long enough to capture the combinatorial intricacies of the problem at hand, such as creating detours towards the goal in the case of future dangerous states, or avoiding dead-ends in a maze.

The local setting formulation makes our architecture a hierarchical method similar to Blaes et al. (2019); Nachum et al. (2018), and allows for solving tasks that are not typical goal-reaching problems, such as the CHASER environment.

## 6 EXPERIMENTS

To validate our hypothesis that embedding planners into neural network architectures leads to better generalization, we consider several procedurally generated environments (from the ProcGen suite Cobbe et al. (2019) and CRASH JEWEL HUNT) with considerable variation between *levels*.

We compare to two baselines: a standard behavior cloning *imitation learning* baseline using a ResNet18 architecture trained with a cross-entropy classification loss on the same dataset as our method; and a reinforcement learning baseline using the PPO algorithm. More details on the training procedure and the hyperparameters can be found in appendix Sec. D.

For the experimental validation, we aim to anwser the following questions:

- Can NAP be trained to perform well as policies in procedurally generated environments?
- Can NAP generalize in a low data regime, i.e. after seeing only few different levels?
- Can we also solve non-goal-reaching environments?

## 6.1 CRASH JEWEL HUNT

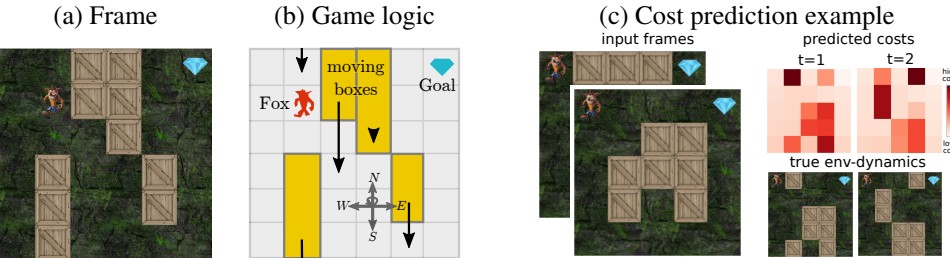

Figure 3: The CRASH JEWEL HUNT environment. The goal for the fox, see (a), is to obtain the jewel in the right most column, while avoiding the moving wooden boxes (arrows in (b)). When the agent collides with a wooden box it instantly fails to solve the task. A prediction of the costs is shown in (c).

We first consider an environment we constructed to test NAP, called CRASH JEWEL HUNT which can be seen in Fig. 3. The environment corresponds to a grid-world of dimensions $h \times w$ where the goal is to move the agent (Fox) from an arbitrary start position in the left-most column to the goal position (jewel) arbitrarily positioned in the right-most column. Between the agent and the goal are obstacles, wooden boxes that move downwards (with cyclic boundary conditions) with velocities that vary across levels but not within a level, see Fig. 3(right). At each time step, the agent can choose to move horizontally or vertically in the grid by one cell or take no action.

To make the task challenging, we sample distinct environment configurations for the training set and the test set, respectively. More concretely, we vary the velocities, sizes and initial positions of the boxes as well as the start and goal positions.

## 6.2 PROCGEN BENCHMARK

In addition to the jewel hunt environment, we evaluate our method on the MAZE, LEAPER and CHASER environments from the ProcGen suite Cobbe et al. (2019). We have chosen these environments because their structure adheres to our assumptions. For the LEAPER we modified the environment such that a grid-world dynamics applies (LEAPER(GRID)). Based on performance of the baselines, the resulting (LEAPER(GRID)) is not an easier environment.

The MAZE and the LEAPER(GRID) tasks have a static goal whose position only varies across levels, whereas the CHASER requires collection of all orbs without contact with the spiders, so the local goals need to be inferred on the fly. The CHASER environment is also **particularly challenging** as even the expert episodes require on average 150 steps, each of which carries a risk of dying. For this reason, we used three expert trajectories per level.

## 6.3 RESULTS

We train our method (NAP) and the imitation learning baseline until saturation on a training set, resulting in virtually $100\%$ success rate when evaluating on train configurations in the environment. For the PPO baseline we use the code from Cobbe et al. (2019) and provide also two subsequent frames and 200M time steps for training. For our method we also report performance of a version with access to the true start and end-point prediction (NAP+ oracle), with the exception of the CHASER where true goals are not well-defined.

In Fig. 4 we show the performance of the methods when exposed to different number of levels at training time. As reported in Cobbe et al. (2019), the baselines have a large generalization gap, and also poor performance when $< 10\,000$ levels are seen. We find that NAP shows strong generalization performance, already for $< 500$ levels. In some environments, such as the MAZE we obtain near $80\%$ success rate already with just 100 levels which is reached by PPO after seeing $200\,000$ levels. For the CRASH JEWEL HUNT $5 \times 5$ already with 30 trajectories a third of the 1000 test-levels can be solved, the baseline manages less than 50 out of the 1000.

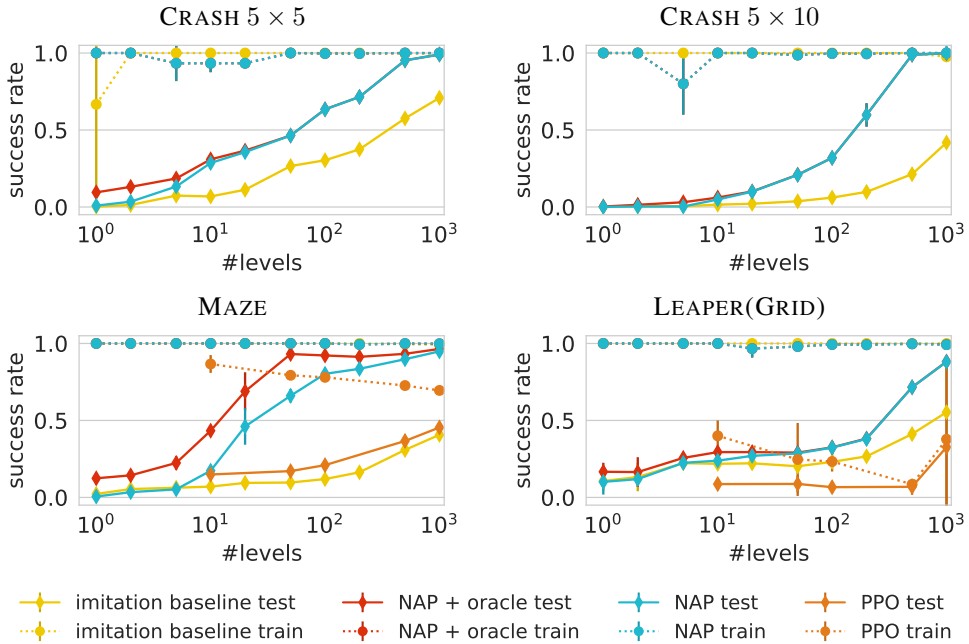

Figure 4: Generalization performance depending on the number of seen levels during training. Dashed lines indicate performance on training-levels and solid lines show the generalization to unseen levels. NAP shows good generalization already after 100 levels in most cases.

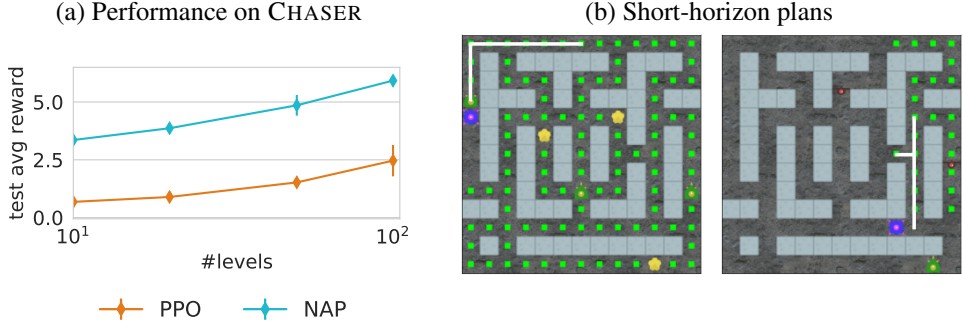

Figure 5: Performance of NAP against PPO on the CHASER environment (a) trained on 10, 20, 50, and 100 levels. In (b) we show the short-horizon plans (white) of the agent (blue) at step 5 and 110 in the environment.

## 6.4 SENSITIVITY TO THE PLANNING HORIZON

We provide a sensitivity analysis of the performance with different planning horizons. Our results indicate that longer horizons benefit environments with increased dynamical interactions. As apparent from Fig. 6, our method outperforms the imitation baseline in the crash environment, the gap between the methods being correlated with the complexity of the environment ($5 \times 5$ vs $5 \times 10$). It can be seen also that making the planning horizon smaller in these environments hurts performance.

On the other hand, for environments with no dynamics, such as the maze environment, there is no benefit in using time-dependent costs, as expected. Nevertheless, there is still strong performance gain in generalization when using NAP oppose to vanilla imitation learning from expert trajectories.

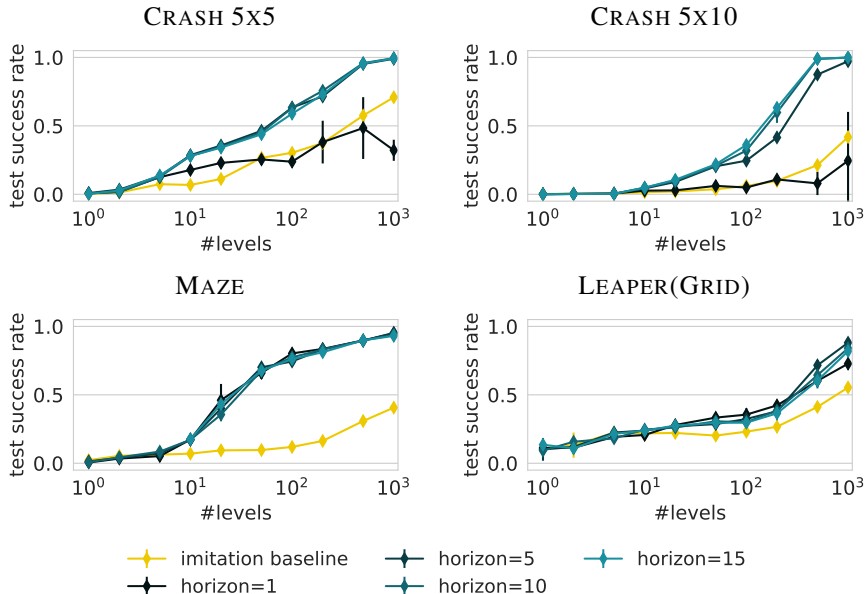

Figure 6: Test success rate of our method with different horizon lengths. The solver assumes that the last horizon step costs remain to infinity. In this sense, the horizon of length 1 corresponds to a static solver.

## 7    DISCUSSION

We have shown that hybrid neuro-algorithmic policies consisting of deep feature extraction and a shortest path solver – made differentiable via blackbox differentiation (Vlastelica et al., 2020) – *enable learning policies that generalize to unseen environment settings in the low-data regime*. Hybrid architectures are a stepping stone towards better use of inductive biases that enable stronger generalization. In NAP, the inductive bias that we impose is the topology of the latent planning graph in conjunction with a planning algorithm. Introducing the shortest-path solver as a module shifts the combinatorial complexity of the planning problem to efficient algorithmic implementations while alleviating the learning of good representations for planning.

Although there is a clear benefit in using NAP, the method comes with certain caveats. We assume that the topological structure (i.e. that there is an underlying grid structure with a set of 5 actions) of the latent planning graph is known a priori. Furthermore, we assume that the structure of the latent graph is fixed and not dynamically changing over time, i.e. that each available action at a vertex corresponds to the same edge. Any results allowing to abandon some of these assumption will vastly increase applicability of this method and should be of immediate interest.

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

## A    DATA GENERATION

In order to do imitation learning, we needed to create expert data. For CRASH $5 \times 5$, CRASH $5 \times 5$, LEAPER(GRID) and MAZE we can determine the exact ground truth costs leading to optimal behavior. As an example, CRASH $5 \times 5$ contains moving boxes that when encountered lead to instant death, meaning infinite costs and otherwise the fixed cost of moving around in the environment.

Since the environments become deterministic for a fixed random seed, we first unrolled their dynamics for each level. After obtaining the underlying grid structure and entities, we labeled them with costs and constructed a graph that reflects the grid structure. An expert trajectory is constructed by applying Dijkstra's algorithm on this graph and the human-labeled costs and then executing in simulation.

For the CRASH JEWEL HUNT experiments, we randomly sampled 2000 solvable levels by varying number of boxes per column, their speed, the agent start position and the jewel position. The training levels were taken from the first half and the second half of levels was used for testing.For the LEAPER(GRID) and MAZE environments we have taken the levels determined by seeds 0-1000.

For CHASER, we applied a similar procedure but additionally, we recorded two sets of human trajectories, as we observed benefits in performance by incorporating more different expert trajectories for the same level. Since both the search procedure and human labeling are time consuming for this environment, we collected fewer expert trajectories for the CHASER than for the other environments, $3 \times 100$, two thirds of which are from human players.

For this reason, we constructed the dataset for the CHASER by recording human trajectories. Moreover, we observed benefits in the training by incorporating more different expert trajectories for the same level.

Levels seeds 1000000-1001000 were taken for testing in the PROCGEN experiments.

## B    ENVIRONMENTS

Our method is applicable in discrete environments, therefore we evaluated on environments from the PROCGEN benchmark and the CRASH JEWEL HUNT environment.

We created the CRASH JEWEL HUNT environment to evaluate our method, where the goal is for the fox (Crash) to reach the jewel. We found this environment convenient since we can influence the combinatorial difficulty directly, which is not true for the PROCGEN benchmark where we are limited to the random seeds used in the OpenAI implementation. The sources of variation in the CRASH JEWEL HUNT are the box velocities, initial positions, sizes, as well as the agent initial position and the jewel end position.

We modified the LEAPER environment to make discrete steps in the world in order to make our method applicable. This involved making the logs on the river move in discrete steps as well as the agent. For an additional description of the PROCGEN environmnets, we refer the reader to Cobbe et al. (2019).

## C    NETWORK ARCHITECTURE AND INPUT

For all of our experiments, we use the PyTorch implementation of the ResNet18 architecture as the base of our model. All of the approaches receive two stacked frames of the two previous time steps as input in order to make dynamics prediction possible. For the PPO baseline, we didn't observe any benefit in adding the stacked frames as input and we used stable-baselines implementation from OpenAI in order to train it on the PROCGEN environments.

In the case of the behavior cloning baseline, the problem is a multi-class classification problem with the output being a multinomial distribution over actions.

For the variant NAP + oracle, we train a cost prediction network on top of which we run the Dijkstra algorithm on the outputted costs of the planning graph. This requires modifications to the original ResNet18 architecture. We remove the linear readout of the original ResNet18 architecture and replace it with the preceded by a convolutional layer of filter size 1 and adaptive max pooling layer to

obtain the desired dimensions of the underlying latent planning graph. More concretely, the output $x$ of the last ResNet18 block is followed by the following operation (as outputted by PyTorch) to obtain the graph costs:

```
Sequential(
            Conv2d(256, 2, kernel_size=(1, 1), stride=(1, 1))
            Abs()
            AdaptiveMaxPool2d(output_size=(grid_height, grid_width))
)
```

Where grid_{height,width} denotes the height and width of the planning grid. For the full variant of NAP with goal and agent position prediction we have a separate position classifier that has the same base architecture as the cost prediction network with 2 additional linear readouts for the likelihoods of the latent graph vertices, more concretely (as outputted by PyTorch):

```
Sequential(
            Conv2d(256, 2, kernel_size=(1, 1), stride=(1, 1))
            Abs()
            AdaptiveMaxPool2d(output_size=(grid_height, grid_width))
            Flatten()
            Linear(grid_height × grid_width, grid_height × grid_width)
)
```

For training the position classifier, we use a standard cross-entropy loss on the likelihoods. For NAP with position classification, we use the ground-truth expert start and goal positions to calculate the Hamming loss of the predicted path by the solver. At evaluation time, NAP uses the position classifier to determine the start and end vertices in the latent planning graph.

## D  TRAINING PROCEDURE

For CRASH $5 \times 5$, CRASH $5 \times 5$, LEAPER(GRID) and MAZE we train the models on the same #levels, namely 1, 2, 5, 10, 20, 50, 100, 200, 500 and 1000. We evaluate on unseen 1000 levels in order to show that NAP exhibits superior generalization. The levels are generated as per description in section A. For each dataset size we run experiments with 3 random seeds and normalize the data to be zero mean and unit variance. For all experiments, we make use of the ADAM optimizer.

We determine the number of epochs for training depending on each dataset size as $\min(150000/\text{\#levels}, 15000)$ to have roughly the same number of gradient updates in each experiment. We take the minimum over the 2 values because for smaller number of levels a large number iterations is not necessary to achieve good performance, but for a larger number of levels it is necessary.

|  | CRASH $5 \times 5$ | CRASH $5 \times 10$ | LEAPER(GRID) | MAZE |
|---|---|---|---|---|
| learning rate | $10^{-3}$ | $10^{-3}$ | $10^{-3}$ | $10^{-3}$ |
| $\alpha$ | 0.2 | 0.2 | 0.15 | 0.15 |
| $\lambda$ | 20 | 20 | 20 | 20 |
| resnet layers | 4 | 4 | 4 | 4 |
| kernel size | 4 | 4 | 6 | 6 |
| batch size | 32 | 32 | 16 | 16 |

Table 1: Training hyperparameters, where $alpha$ denotes the margin that was used on the vertex costs and $\lambda$ the interpolation parameter for blackbox differentiation od Dijkstra. We vary the kernel size of the initial convolution for ResNet18.

For the CHASER, the training conditions were analogous to the other environments, only of slightly smaller scale due to its higher complexity. Models were trained on 10, 20, 50, and 100 levels and evaluated on 200 unseen levels. Each model trained for 40 epochs.

|  | CHASER |
|---|---|
| learning rate | $1e^{-3}$ |
| $\alpha$ | 0.2 |
| $\lambda$ | 40 |
| resnet layers | 3 |
| kernel size | 4 |
| batch size | 16 |

Table 2: Training hyperparameters for the CHASER experiment, where $alpha$ denotes the margin that was used on the vertex costs and $\lambda$ the interpolation parameter for blackbox differentiation od Dijkstra.

### D.1 PPO TRAINING PROCEDURE

The training of the PPO baseline is exactly the same as described in Cobbe et al. (2019) using the official code from `https://github.com/openai/train-procgen`, see Table 3 for the used parameters. The network architecture is the IMPALA-CNN. The algorithm is trained on the specified number of levels for 200 million environments interactions. We report numbers for 5 independent restarts.

| learning rate | $5e^{-4}$ |
|---|---|
| $\alpha$ | 0.2 |
| discount $\gamma$ | 0.999 |
| entropy coefficient | 0.01 |
| steps per update | $2^{16}$ |

Table 3: PPO hyperparameters, as used in Cobbe et al. (2019).

