# OpenReview forum: "Neuro-algorithmic Policies for Discrete Planning"
_ICLR.cc/2021/Conference — Reject_

### Official Review · AnonReviewer1 · 2020-10-21
**Good paper with solid set of experiments to back it up**

**Rating:** 7
**Confidence:** 3

**Review:**

This paper proposes a policy architecture that embeds graph search
within it. The proposed architecture is suggested to show strong
generalization capabilities due to the embedded combinatorial/discrete
search. The authors study this architecture in the context of
imitation learning from a small number of expert demonstrations. The
core approach builds on a previous work (Vlastelica et al. 2020) that
provides a framework for embedding combinatorial search within a
differentiable model.

Overall, I think this is a very solid work, and I recommend
acceptance. The problem being studied is eminently important, and
right now the field is in great need of policy optimization methods
that leverage the power of discrete search algorithms. The experiments
are quite comprehensive, encompassing various gridworlds, and
generally addressing the baselines I would have expected to see (most
importantly, Figure 6 which shows the performance vs. amount of data).

The biggest concern I have with the paper is that it is extremely
unclear what exactly is meant by the assumption of a known topology of
the gcMDP. Does this topology basically represent the transition
function (i.e., edges in the discrete state graph)? If so, then it
seems unfair to compare to IL and RL baselines that are not given
access to this information. If not, then I really hope the authors can
clearly convey (perhaps through a small example) what the topology is,
and how much practical engineering effort is required to create it in
your domains.

I am also curious how much stochasticity in the environment can be
handled by your method. This is in reference to the sentence "Although
the actual gcMDP solved by the expert may be stochastic, we learn a
deterministic latent approximate gcMDP".

One smaller question: when you say "In gcMDPs the reward is such that
the maximal return can be achieved by reaching the goal state g" --
this feels like a pretty weak condition to me, because it can be
achieved by simply making the reward at g higher than the returns of
any trajectory not reaching g. Is this condition necessary and
sufficient for your approach to make sense?

Looking ahead, I am curious if the authors have considered extending
the proposed system to continuous environments (e.g., if states are
images)? Perhaps a path forward here would be to learn an embedding to
a discrete latent space, and then apply NAP. I guess the challenge
would be that you would not be able to know the topology of that
latent a priori.

---

> ### Author Response · Authors · 2020-11-19
> **Answer to Reviewer 1**
>
> Thank you for the comments and reading our paper.
>
> > "In gcMDPs the reward is such that the maximal return can be achieved by reaching the goal state g"
>
> Goal-reaching tasks are quite common in RL, see for instance papers using Hindsight experience replay. We make the requirement explicit that a goal needs to be reached. This makes the shortest path a viable solution to the restricted MDP setting.
>
> > “The biggest concern I have with the paper is that it is extremely unclear what exactly is meant by the assumption of a known topology of the gcMDP”
>
> The topology describes what kind of structure the underlying MDP has, i.e. which state transitions can be made. In the case of gridworld environment, the topology of the MDP is such that transitions can be made only to 4 neighboring vertices (except on the edges of the grid). We clarify this in more detail in the paper.”
>
> > “I am also curious how much stochasticity in the environment can be handled by your method. This is in reference to the sentence 'Although the actual gcMDP solved by the expert may be stochastic, we learn a deterministic latent approximate gcMDP'".
>
> This is a good question that we want to address in future work. We believe that the costs would account for stochasticities rendering the found shortest path the one taking potential perturbations into account (as the expert performs well on average). Replanning in the MPC framework should make the policy locally resilient.
> So we don’t consider this as a limitation. Additionally, most real-world use cases are indeed deterministic, the stochastic formulation is used to account for unknown factors of variation that change the transition dynamics.
>
> > “Looking ahead, I am curious if the authors have considered extending the proposed system to continuous environments”
>
> Yes, this is indeed something we plan to consider for future work.

---

### Official Review · AnonReviewer3 · 2020-10-25
**Review of NAP**

**Rating:** 3
**Confidence:** 4

**Review:**

Summary: The paper studies the problem of image-based planning in discrete state/action spaces. The paper proposes a neuro-algorithmic policy that can be trained end-to-end by differentiation and used together with a shortest path solver.

Strengths:

i) The motivation, organization and the overall writing of the paper are clear.

Weakness:

i) The manuscript is missing very relevant pieces of works that should have been discussed in detail and included as baselines in the experimental results section. Namely, the previous work (i.e., Latplan) [1,2] that performs (classical) planning from images in latent spaces using variational autoencoders in combination with off-the-shelf automated planners. Similarly, experimental comparison to work [3] is missing.

ii) How does the proposed approach (i.e., NAP) reason about the value of the planning horizon T which is an important aspect of automated planning? Note that Latplan would again provide useful insights here (since it leverages off-the-shelf automated planners).

iii) The benchmark domains seem too simple. For a more realistic image-based task, see Meta-World [4].

Additional comments:

Page 1: In the third paragraph, can you please ground the claim for the statement “transition model is often harder than learning a policy”? Moreover, instead of just saying “there exists several approaches…”, you should include previous works [5,6,7] as references that learn such transition models for planning.

Page 2: Second item in the list on page 2: generalizing policies -> generalized policies

References:
[1] Classical Planning in Deep Latent Space: Bridging the Subsymbolic-Symbolic Boundary, Asai and Fukunaga AAAI-18.

[2] Learning Neural-Symbolic Descriptive Planning Models via Cube-Space Priors: The Voyage Home (to STRIPS), Asai and Muise IJCAI-20.

[3] Learning latent dynamics for planning from pixels, Hafner et al. ICML-19.

[4] A benchmark and evaluation for multi-task and meta reinforcement learning, You et al. CoRL 2019.

[5] Nonlinear Hybrid Planning with Deep Net Learned Transition Models and Mixed-Integer Linear Programming, Say et al., IJCAI-17.

[6] Scalable Planning with Deep Neural Network Learned Transition Models, Wu et al. JAIR.

[7] Optimal Control Via Neural Networks: A Convex Approach, Chen et al., ICLR 2019.

---

> ### Author Response · Authors · 2020-11-19
> **Answer to Reviewer 3**
>
> Thank you for your review and comments on our paper.
>
> We have included all of the references that you mentioned + more in the related work section to better relate our work to other approaches.
>
> > " How does the proposed approach (i.e., NAP) reason about the value of the planning horizon T which is an important aspect of automated planning? Note that Latplan would again provide useful insights here (since it leverages off-the-shelf automated planners)."
>
> Automating the horizon length is an independent problem and any solution for it is composable with our architecture. The main difference to LatPlan is that we have an end-to-end differentiable hybrid architecture, whereas LatPlan learns each individual component separately.
>
> On the question of an additional more realistic benchmark such as Meta-World - we believe that the current benchmarks that we propose are enough to show the key hypothesis of the paper, that neuro-algorithmic policies enable strong generalization through combinatorial inductive biases. Tackling more realistic benchmarks we leave for future work. Meta-world concretely was designed for benchmarking meta-learning algorithms, we propose here an offline imitation learning method. We have stated this more explicitly in the introduction, conclusion and related work.
>
> We have provided additional clarifications in the text regarding the mentioned points and have fixed the typos.

---

### Official Review · AnonReviewer2 · 2020-10-28
**Review [Updated]**

**Rating:** 3
**Confidence:** 4

**Review:**

**SUMMARY**

This work proposes a novel neuro-algorithmic policy architecture for solving discrete planning tasks. It takes a high-dimensional image input and processes it through modified ResNet encoders to obtain a graph cost map and a start/goal heatmap. This is fed into a differentiable Dijkstra algorithm to obtain the shortest trajectory prediction which is trained using an expert-annotated trajectory via a Hamming distance loss. This module is evaluated in two dynamic game environments demonstrating generalization to unseen scenes.


**STRENGTHS**
- The general idea of integrating BlackBox combinatoric shortest path algorithms in a differentiable planning module is interesting and has a lot of potential to be useful.

**WEAKNESSES**
- The novelty compared to Vlastelica et al. (2020) is not clear.
- The design choices are not clearly justified and the considered use-cases for this particular architecture are limited.
- Important information is missing to make this work reproducible (see reproducibility section)
- The evaluation considers only shortest-path planning scenarios that are amenable to the proposed architecture (see evaluation section). The authors should either provide a clear motivation of the considered scenario types or evaluation on scenarios learning more complex representations.

**CLARITY**

The general idea of the work is clearly written although important information for reproducibility is missing (see below). I also felt that the authors did not make particularly good use of space. For example, Sec. 3 and Sec 4.1 could be condensed into a joint background section, leaving more space for more detailed experiments. The information in Sec 4.3 seems to be a better fit for either the related work or the conclusion section.

Smaller clarification questions:
- What is the y axis in fig 5 (a).
- In the conclusion: What exactly is meant by knowing the topological structure of the latent planning graph a priori? How is this incorporated as an inductive bias into the neural network?

**REPRODUCIBILITY**

The results in this work are not reproducible. Relevant information on training (e.g. which optimizer was used? what were the learning rates? ...), hyperparameters (which parameters were tuned? which range was considered? how were they tuned?), baseline method training (e.g., how long was PPO trained, how exactly were rewards defined, ...) and environment generation settings are missing.


**EVALUATION**

The evaluation seems one of the weak points of this work. The problems here are threefold:
1. the number of considered tasks is very limited and their type is very limited to scenarios where one image input provides enough information to generate a full cost map. This does not hold in most planning tasks.
2. Because the architecture itself is a claimed contribution, this work would require a much more thorough evaluation of architecture design decisions such as which underlying CNN is used, what is the
3. Simply using the PPO baseline is insufficient. First, there is no discussion on how and why this algorithm was chosen as a baseline. Second, more recent or closer related baselines are missing. Some of those are mentioned in related work and they seem to be more fair/useful comparison methods.


**NOVELTY / IMPACT**

The work is not sufficiently motivated. While planning, in general, is an important problem and differentiable planners are an important research topic, the motivation of this work is not clear. The authors should not only name the use-cases where an intermediate planning module might be beneficial but also discuss what the main insights of this work are. As the authors write themselves, a differentiable implementation of TDSP in a neural network can simply be achieved by applying theory from Vlastelica et al. (2020). In fact, that paper already demonstrates the use of Dijkstra in a neural network computation graph. This opens up the question of what is the key idea and value of the present work?

Furthermore, the authors did not sufficiently consider/discuss existing related work. The related work does not sufficiently cover state-of-the-art. Combining planning modules with deep learning pipelines (although in slightly different ways) has been considered in several works. More importantly, the related work should discuss why the proposed approach is beneficial compared to approaches containing differentiable planners. The related work is also missing numerous relevant papers. Some potential examples involve:

- Kuo et al., Deep sequential models for sampling-based planning, IROS 2018
- Kumar et al., LEGO: Leveraging Experience in Roadmap Generation for Sampling-Based Planning, IROS 2019
- Gupta et al., Cognitive mapping and planning for visual navigation, CVPR 2017
- Savinov et al., Semi-Parametric Topological Memory for Navigation, ICLR 2018
- Tamar et al., Value Iteration Networks, NeurIPS 2016

Finally, the related work section should not only provide a broader discussion of related works but also more clearly emphasize the relationship between these works and the present work (e.g. how and in what context is the present work better more useful than related works? Which ideas are used from related work and what is new to this work? ...).

**REVIEW SUMMARY**

Overall, I recommend rejection of this work because the issues raised in this review cannot be resolved without a significant amount of work. While this may be discouraging, I believe that the authors are considering an interesting topic that has a lot of potential and, after resolving the issues raised in this review, may result in a successful submission.

**POST-DISCUSSION SUMMARY**
I want to thank the authors for answering my questions and correcting misunderstandings and updating the paper. I still recommend rejection of this work given that the novelty is not yet fully clear. While the updated list of contributions is indeed a better match for this work, claiming that correct inductive biases improve generalization does not seem to be a new insight. As mentioned in my initial review, I still believe that the general line of work has a lot of potential and want to encourage the authors to fully address the general concerns raised in the reviews and resubmit the work.

---

> ### Author Response · Authors · 2020-11-19
> **Answer to Reviewer 2**
>
> Thank you for taking the time to read our paper and for the constructive feedback.
>
> > “This module is evaluated in two dynamic game environments demonstrating generalization to unseen scenes”
>
> We evaluate on 4 different dynamic game environments (3 if we count the crash environments as 1)  and 1 static (the maze environment). The generalization is with respect to unseen levels.
>
> > “The evaluation considers only shortest-path planning scenarios that are amenable to the proposed architecture”
>
>  Indeed, we see this as one of the strengths of model-based and hybrid approaches, this allows certain structural priors to be used for encoding knowledge of the problems at hand in order to solve them more efficiently, similarly as stated in [1]. We make this more explicit in the introduction.
>
> We acknowledge that there are problems of reproducibility, we have made adjustments to the writeup and published the code. Please read the general answer.
>
> Regarding novelty concerns, please read the general answer.
>
> >“In the conclusion: What exactly is meant by knowing the topological structure of the latent planning graph a priori? How is this incorporated as an inductive bias into the neural network?”
>
> The network outputs a DAG with a specific topology (encoding that there are 5 actions) and we learn the costs of the edges, this is the form of the inductive bias that we impose on the policy, we make this more explicit in the conclusion.
>
> Regarding evaluation concerns - We’ve updated in detail the information about the architecture and the hyperparameters used for evaluation, this can be found in the appendix. PPO was chosen as a baseline because it was used in the original ProcGen paper [2]. Note that PPO is indeed unfair for different reasons and it mainly serves to show that learning tasks that require combinatorial generalization is too hard to simply apply model-free RL algorithms. We will clarify this in the appendix.
>
> Our method is an offline imitation learning method, which means that we don’t get any additional information by interacting with the environment. This is the reason we have chosen to compare to a behavior cloning method (we’ve made this more explicit in the writeup), since baselines such as GAIL and SQIL are online, i.e. they combine expert data with newly sampled data (as R4 noted). A more fair comparison would be an offline RL method, such as CQL [3] (Conservative Q-Learning), but this method has only been proposed recently, before the submission of this paper, and hasn’t been published yet. Note that offline RL methods are solving a bit of a different problem by also trying to learn policies that are better than sub-optimal expert policies, since we propose a specific hybrid architecture that is a closed-loop policy, it could be also trained via an offline RL method, which is a promising area of future work. We’ve stated this connection in the related work section more explicitly (which we have updated considerably).
>
> We have included all of the references that you suggested + more to better put our work in perspective, please read the newly updated related work. This hopefully also makes the motivation of the work clearer.
>
>
> [1] Karkus, Peter, et al. "Differentiable algorithm networks for composable robot learning." RSS 2019.
>
> [2] Cobbe, Karl, et al. "Leveraging procedural generation to benchmark reinforcement learning." arXiv preprint arXiv:1912.01588 (2019).
>
> [3] Kumar, Aviral, et al. "Conservative Q-Learning for Offline Reinforcement Learning." arXiv preprint arXiv:2006.04779 (2020).

---

### Official Review · AnonReviewer4 · 2020-10-29

**Rating:** 4
**Confidence:** 4

**Review:**

#### Summary:
This paper presents a method to train a neural network to predict the time-dependent costs, and start and goal states needed to run time-dependent shortest-path planning in a dynamic 2-D environment. The non-differentiability of the path planning is handled by recent work on differentiating through blackbox combinatorial solvers from [1]. The method is trained in a supervised manner from expert trajectories. Evaluations are presented on 2-D time-varying games where the addition of the path-planner is shown to improve performance over an imitation learning and PPO baseline.

##### Pros:
- The method is simple and looks to perform well in the experiments.
- The description of the approach is fairly clear and well written.

##### Cons:
- The novelty relative to [1] is quite low. Specifically, [1] already uses the blackbox differentiation to do shortest path solving using a neural network architecture to estimate the costs.
- Overall, the evaluation is a bit lacking.
- The paper is missing key details that would aid anyone trying to build off of or replicate this work.

#### Decision:
Given the low novelty and not-so-strong evaluation, I do not recommend this paper for acceptance.

#### Questions:
1. As far as I can tell, the main novelty relative to [1] is the use of a time-varying cost function (albeit encoded into a shortest-path search as in [1]) and the learned start and goal encoder. Am I missing something else?
2. How does this approach compare to the recent work of [2]?
3. Is the imitation learning baseline behavior cloning or something else? This needs more detail.
4. The set of baselines is insufficient since PPO does not make use of the expert trajectories. Something like GAIL [3] or SQIL [4], which combines RL and imitation learning, should be used in addition to the other two baselines.
4. In Section 4.2, the set of admissible paths seems like it may be a hard constraint to satisfy. How do you accomplish this?

#### Comments:
1. Given that there don’t seem to be great baselines for these tasks, it would improve the evaluation to also show results on a simpler task that does have other baselines, such as non-time-varying shortest path search.
2. Overall, the paper could use more detail on the architecture, the hyperparameters, the baselines, the domains, and anything else needed to get this to work. It would be quite nontrivial to reproduce this work from the paper alone. A subsection on training would also be nice, as the training details seem to be scattered throughout the other sections, which makes it hard to piece together exactly how it’s done.
3. Please learn the difference between citep and citet (from natbib) and use them appropriately to make the citations more intelligible for the reader.
4. The first paragraph of the intro doesn’t make it clear that this work falls on the first end of the discussed spectrum and uses expert demonstrations to learn from.
5. For the related work section, don’t just list related work, contrast that work to your own so that the reader gets a better understanding of your method in the context of the existing work.
6. Typo: preceding horizon planning → receding horizon planning.


[1] Vlastelica, Paulus, Musil, Martius, Rolinek. Differentiation of Blackbox Combinatorial Solvers (2020).
[2] Yonetani, Taniai, Barekatain, Nishimura, Kanezaki. Path Planning using Neural A* Search (2020).
[3] Ho, Ermon. Generative Adversarial Imitation Learning (2016).
[4] Reddy, Dragan, Levine. SQIL: Imitation Learning via Reinforcement Learning with Sparse Rewards (2019).



****************************

Thank you for the response and updates to the paper. Given the number of changes required, I encourage the authors to resubmit elsewhere with the updated paper, ideally with additional experimental comparisons as discussed. Note that there are many other offline RL works, such as BCQ or BEAR, that you could use (see, e.g., "D4RL: Datasets for Deep Data-Driven Reinforcement Learning" or "RL Unplugged: Benchmarks for Offline Reinforcement Learning" for relevant algorithms).

---

> ### Author Response · Authors · 2020-11-19
> **Answer to Revewer 4**
>
> Thank you for your review and comments on our paper.
>
>  Q 1: Please see the general answer
>
> Q 2: We agree that [2] adresses a similar problem and could potentially be modified to the RL setups we consider. However, please note that [2] appeared on ArXiv in the final weeks before the submission deadline. Also, the code has not been made available (in fact, we even requested it from authors) making any direct comparisons impossible. To state the differences, [2] does not solve a control problem (i.e. does not provide a closed-loop policy), does not deal with goal selection, nor looks at the low-data regime generalization. We expanded our related work section to address this.
>
> Q 3+4: Our method is an offline imitation learning method, which means that we don’t get any additional information by interacting with the environment. This is the reason we have chosen to compare to a behavior cloning method (we’ve made this more explicit in the writeup), since baselines such as GAIL and SQIL are online, i.e. they combine expert data with newly sampled data (as R4 noted). A more fair comparison would be an offline RL method, such as CQL [X] (Conservative Q-Learning), but this method has only been proposed recently, before the submission of this paper, and hasn’t been published yet. Note that offline RL methods are solving a bit of a different problem by also trying to learn policies that are better than sub-optimal expert policies, since we propose a specific hybrid architecture that is a closed-loop policy, it could be also trained via an offline RL method, which is a promising area of future work. We’ve stated this connection in the related work section more explicitly (which we have updated considerably).
>
> Q 5: The set of admissible paths here only means actually connected paths without loops. This is something that the shortest path solver is enforcing automatically and is one of the benefits of using an embedded solver. There are no additional constraints needed. We clarified this more explicitly in the paper.
>
> C1 : The maze task is exactly this kind of task where there is no dynamics. We show this in the sensitivity analysis of the horizon length in Fig. 6. The horizon length of H=1 (which corresponds to a static solver) is enough to learn the task sufficiently. Also, it is clear for the other tasks that a bigger horizon length is beneficial for performance. We’ve clarified this more in the figure caption.
>
> C2: We have uploaded the training code and also described the training procedure and the model architecture in the supplementary.
>
> C3: We’ve fixed all of the misuses of \citep vs \citet
>
> C4: We’ve stated that our method is an offline imitation learning method more explicitly in the introduction.
>
> C5: We have updated the related work section considerably and stated explicitly the difference to other approaches.
>
> [X] Kumar, Aviral, et al. "Conservative Q-Learning for Offline Reinforcement Learning." arXiv preprint arXiv:2006.04779 (2020).

---

### Decision · Program_Chairs · 2021-01-07
**Final Decision**

**Decision:**

Reject

**Comment:**

This paper is about training a discrete policy that maps an image representation through a differentiable time-dependent path planning module. The method is based on [1] and the reviewers are concerned about lack of novelty with respect to this work, and also with [2], however the latter only appeared a few weeks before the ICLR deadline, so I am not factoring it in my recommendation. Unfortunately, in light of [1], 3/4 reviewers do not recommend acceptance, and I agree with them.

[1] Vlastelica, Paulus, Musil, Martius, Rolinek. Differentiation of Blackbox Combinatorial Solvers (2020).
[2] Yonetani, Taniai, Barekatain, Nishimura, Kanezaki. Path Planning using Neural A* Search (2020).